# Analysis and Optimization of Dynamic and Static Characteristics of the Compliant-Amplifying Mechanisms

**DOI:** 10.3390/mi14081502

**Published:** 2023-07-26

**Authors:** Jin Wang, Zijian Jing, Zongliang Xie, Zongqi Ning, Bo Qi

**Affiliations:** 1Key Laboratory of Optical Engineering, Chinese Academy of Sciences, Chengdu 610209, China; 15735044578@163.com (J.W.); zijianjing@foxmail.com (Z.J.);; 2National Key Laboratory of Optical Field Manipulation Science and Technology, Chinese Academy of Sciences, Chengdu 610209, China; 3Institute of Optics and Electronics, Chinese Academy of Sciences, Chengdu 610209, China; 4University of Chinese Academy of Sciences, Beijing 100049, China

**Keywords:** compliant-amplifying mechanisms, multi-body system transfer matrix method, displacement amplification ratio, resonance frequency, iterative optimization algorithm

## Abstract

Compliant amplifying mechanisms are used widely in high-precision instruments driven by piezoelectric actuators, and the dynamic and static characteristics of these mechanisms are closely related to instrument performance. Although the majority of existing research has focused on analysis of their static characteristics, the dynamic characteristics of the mechanisms affect their response speeds directly. Therefore, this paper proposes a comprehensive theoretical model of compliant-amplifying mechanisms based on the multi-body system transfer matrix method to analyze the dynamic and static characteristics of these mechanisms. The effects of the main amplifying mechanism parameters on the displacement amplification ratio and the resonance frequency are analyzed comprehensively using the control variable method. An iterative optimization algorithm is also used to obtain specific parameters that meet the design requirements. Finally, simulation analyses and experimental verification tests are performed. The results indicate the feasibility of using the proposed theoretical compliant-amplifying mechanism model to describe the mechanism’s dynamic and static characteristics, which represents a significant contribution to the design and optimization of compliant-amplifying mechanisms.

## 1. Introduction

Benefiting from advantages that include high resolution, fast response times, and high bandwidths [1,2,3], piezoelectric actuators (PZAs) are used widely in high-precision instruments for applications in aerospace [4,5,6,7], precision machining [8,9,10], semiconductor technology [11,12], and other fields [13,14,15,16,17]. To allow the advantages offered by PZAs to be used fully, the compliant-amplifying mechanism, which plays an important role in piezoelectric stroke amplification, has attracted considerable attention in recent years. Given that the finite element method (FEM) fails to reflect the internal mechanism or the effects of the geometric parameters of flexible amplifying mechanisms, it has thus become necessary to develop a suitable model that is both static and dynamic.

Investigations have shown that research on static modeling of the compliant-amplifying mechanism has attracted a great deal of attention, and the methods used [18] mainly include Castigliano’s second theorem [2,19,20,21,22], the flexibility matrix method [23,24,25,26,27,28], the matrix displacement method [1,29,30], the elastic beam theory [31,32,33,34,35,36,37], and kinematic principles [35,38,39,40,41,42]. Based on these methods, the effects of the main parameters of the mechanism on its static characteristics have been analyzed, and the relationship between the input and output displacements of the mechanism was established. Dynamic modeling methods have mostly involved use of the Lagrange Equation [22,24,27,42,43,44,45], and part of the analysis was performed according to d’Alembert’s principle in combination with the matrix displacement method [1,30]. Further analysis showed that most of the static analysis efforts have focused on modeling of the flexible bridge arm and modeling of the piezoelectric actuators, and the effects of their pseudo-rigid units on the properties of the mechanism have not been considered fully. The dynamic modeling analyses are mainly concerned with analysis of the mechanism carrier, while no specific analysis has been performed to address the effects of the main geometric parameters on the mechanism. More importantly, there are very few unified analysis models of compliant-amplifying mechanisms that include both dynamic and static analyses.

The transfer matrix method for multibody systems (MSTMM) is a new method proposed by Rui et al. to study the dynamics of multibody systems by using a transfer matrix [46]. Its system matrix has a low order and does not need to consider the overall dynamic equations of the system. It has the characteristics of flexible modeling, simple and efficient analysis, fast calculation speed, easy operation and programming, and is widely used in industry product development and experimental dynamic design [47,48,49,50].

Therefore, the work presented in this paper established a theoretical mathematical model of the amplifying mechanism using MSTMM. The effects of the main parameters of the mechanism (including the thickness, angle, and width of the bridge arm, the length, width, and height of the pseudo-rigid body units, and the material properties of the mechanism) on its dynamic and static characteristics were analyzed using the control variable method. Furthermore, an iterative optimization algorithm was used to obtain specific parameters that meet the mechanism’s design requirements.

The rest of this paper is organized as follows. In Section 2, the process in which the mathematical model of the amplifying mechanism is established using the transfer matrix method is described. In Section 3, the mechanism’s static characteristics are analyzed. In Section 4, the mechanism’s dynamic characteristics are analyzed. In Section 5, a modified transfer matrix is established to reduce the error caused by the stress concentration. In Section 6, the process used to determine the optimal values that meet the design requirements of the mechanism is introduced. In Section 7, the proposed model is simulated and the mechanism is designed and verified experimentally. In Section 8, the proposed method is applied to practical situations and experiments are performed. Conclusions about the study are presented in Section 9.

## 2. The Mathematical Models Based on MSTMM

Common basic compliant-amplifying mechanism models include both rhombus-type and bridge-type compliant mechanism models. The rhombus-type compliant mechanism is the simplest model in terms of the triangular amplification principle, offering high operating frequencies and strong load capacities [51]. The bridge-type compliant mechanism can be regarded as a rhombus-type compliant mechanism with compact structure and large displacement amplification ratio, so it will not be discussed below in this paper. Schematic diagrams showing the main geometric parameters of the rhombus-type compliant mechanisms are presented in Figure 1. The flexible mechanism is driven using a piezoelectric actuator. When this piezoelectric actuator is elongated laterally, the compliant mechanism produces longitudinal deformation and an output displacement under the action of an input force/displacement.

Using the MSTMM, the rhombus-type compliant mechanisms can be divided into units, as shown in Figure 2a. The rhombus-type compliant mechanism is divided into pseudo-rigid units 1, 3, 5, 7, 9, and 11, flexible units 2, 4, 6, and 8, and piezoelectric unit 10. Meanwhile, the local coordinate system corresponding to each unit is also shown in the figure.

In Figure 2a, $i_{1}$ is the input point of the i-th unit (i=1∼11,i≠5,7), and $i_{i,1}$ and $i_{i,2}$ are the first and second input points of the i-th unit (*i* = 5,7), respectively. The local coordinate system in the area where the unit is located is established by using the input point as the origin. The local coordinate system $x_{i}o_{i}y_{i}\left(i=2,4,6,8\right)$ is inconsistent with the global coordinate directions, and the corresponding coordinate changes should be made during the calculation.

When the transmission route of the system is established, it can be seen that the amplifying mechanism becomes a bifurcation mechanism, and its transmission route is not unique. Figure 2b only shows one interrelationships among the units of the mechanism.

Based on consideration of the need for both static and dynamic analysis of the mechanism, the extended transfer matrix should be established. With reference to [46], the equation for the transfer matrix of the piezoelectric actuator was derived, and the transfer matrix for the piezoelectric actuator in the longitudinal direction is:(1)$$U\left(L\right)=\left[\begin{array}{c} \cos\lambda L\;-\frac{1}{\lambda EA}\sin\lambda L\\ \lambda EA\sin\lambda L\;\cos\lambda L \end{array}\right]$$

The above equation for the piezoelectric actuator’s longitudinal vibration is the same as that for the mass Euler beam. At the same time, when the lateral vibration of the piezoelectric actuator is considered, it can be regarded as a mass Euler–Bernoulli beam with ordinary elastic deformation. Therefore, the piezoelectric actuator can be regarded as a mass Euler–Bernoulli beam as a whole and the transfer matrix is the same as that for the beam. Addition of piezoelectric motion and force characteristics to the equation allows the total transfer expansion matrix of the piezoelectric actuator to be obtained as follows:(2)$$U_{piezoelectric\;}=\left[\begin{array}{c} U_{Euler-Bernoulli}\;XF\\ 0\;\;\;\;1 \end{array}\right]$$
where
(3)$$XF={\left[u_{11}\;0\;0\;0\;u_{15}\;0\right]}^{T}$$

$u_{11}=d_{33}nU_{PZT}$, $u_{15}=F_{PZT}$, and $d_{33}$ is the piezoelectric deformation coefficient. In addition, *n* is the number of piezoelectric ceramic pieces and $U_{PZT}$ is the driving voltage.

Then, based on the sub-transfer equation, the total transfer equation for the amplifying mechanism is established as shown.
(4)$$U_{all}Z_{all}=0$$

Finally, the equation processing is performed.

## 3. Static Characteristic Analysis

The displacement amplification ratio is the main static characteristic of the compliant-amplifying mechanism. To solve it, the compliant mechanism should be in a quasi-static condition, that is, the frequency ω=0 or $\lim_{x\to 0}\omega =0$.

By substituting for the appropriate parameters, the displacement amplification ratio of the mechanism can be calculated based on the mathematical model established using the MSTMM. Figure 3 shows the output displacement characteristics of the amplifying mechanism versus the different voltages applied to the piezoelectric actuator as determined using the MSTMM and the FEM. The slope of the curve $K_{1}=\frac{x_{output}}{U_{input}}=\frac{R_{amp}}{d_{33}n}$ indicates that the static characteristics of the amplifying mechanism are unaffected by changes in the input displacement [25,39]. These two curves are obviously similar, and their variation trends are also comparable, which indicates that the mathematical model of the amplifying mechanism established based on the MSTMM can feasibly be used to describe the mechanism’s static characteristics.

(1)Flexible unit

To study the amplification ratio characteristics of the amplifying mechanism in greater depth, the effects of the different parameters on the output displacement amplification ratio of the mechanism are determined via the method of controlling variables. Figure 4a shows the relationship between the rhombus-type compliant mechanism’s amplification ratio and the bridge arm angle when the FEM results are compared with the MSTMM results. The figure shows that the two methods have similar trends, in that the amplification ratio increases initially and subsequently decreases with increasing bridge arm angle, which is consistent with the previous description in the literature [52]. Furthermore, the relative error between the two models decreases sharply as the angle increases, before gradually stabilizing after 15${}^{\circ }$. Figure 4b shows that the amplification ratio of the rhombus-type mechanism decreases as the thickness of the bridge arm increases, but also shows that the ratio is not affected by the width of the bridge arm, which is consistent with the behavior described in Qi’s article [35]. The different colors in the figure without specific explanation correspond to different Z-axis values. Furthermore, the following explanation will not be provided.

(2)Pseudo-rigid body element

Unlike many other reports in the literature, which focused on studies of the bridge arm, this paper also studies the effects of the parameters of the pseudo-rigid body units on the mechanism’s amplification ratio. Figure 5 shows that the length and the height of the input rigid body unit and the height of the output rigid body do not affect the mechanism’s amplification ratio. The only parameter that affects the amplification ratio is the length of the bridge arm. The amplification ratio is obviously affected when the bridge arm length is changed. When the length of the output rigid body increases, the length of the bridge arm becomes shorter, and the amplification ratio of the mechanism decreases.

## 4. Dynamic Characteristics Analysis

During the dynamic analysis, the frequency ω≠0. The piezoelectric actuator is set to apply an excitation force to the compliant mechanism, where $F_{PZT}=Bcos\left(\omega \right)$. The solution to this equation must therefore be obtained.

### 4.1. Resonance Frequency

Substitution of the parameters for the rhombus-type compliant mechanism allows the resonance frequency results for the mechanism to be calculated here. Using the thickness, width, and angle of the model bridge arm as the variables, the effects of the main geometric parameters on the dynamic characteristics of the rhombus-type compliant mechanism are discussed here.

(1)Flexible unit

Figure 6a shows that the first-order resonance frequency of the mechanism increases as the bridge arm thickness increases, and the error between the MSTMM and FEM results also increases accordingly. Figure 6b shows that the first-order resonance frequency decreases as the arm angle of the mechanism increases. Unlike the static properties, there is no longer a peak point that varies with the angle. A smaller angle corresponds to a larger error in this case. The MSTMM curve in Figure 6c shows that the first-order resonance frequency of the mechanism is not affected by the arm width, although the FEM curve drops slightly. The reason for this phenomenon is that this model is only designed for the dimensions on a plane.

(2)Pseudo-rigid body element

Figure 7 shows the effects of the pseudo-rigid body unit parameters on the resonance frequency of the mechanism. The resonance frequency of the mechanism is shown to decrease as the length of the input rigid body unit and the height of the output rigid body increase, but the frequency increases as the length of the output rigid body unit increases.

### 4.2. Frequency Response Analysis

Substitution of the geometric parameters allows the frequency response of the mechanism in all directions to be calculated here. Figure 8 shows the frequency response results for the rhombus-type compliant mechanism in the main direction in the plane. The FEM simulation results adopt the shaking table experiment method, and the MSTMM calculation results adopt the piezoelectric self-excitation method. It can be seen that the positions of the resonance points and the resonance curve trends are similar, which again verifies the accuracy of the proposed method.

## 5. Hybrid Compliance Improvement Model

Based on analysis of the previous results, it is feasible to use the MSTMM to describe both the dynamic and static characteristics of the compliant mechanism. However, there is a certain error between the established mathematical model and the finite element model, and at small angles in particular, the gap between these models is especially obvious. This error increases when the thickness of the bridge arm increases and when the angle of the bridge arm decreases.

Through a combination of analysis and comparison, it was found that the main cause of this phenomenon is the stress concentration effect in the connection area between the bridge arm of the compliant-amplifying mechanism and the rigid body unit (see Figure 9a), which results in uneven distribution of the deformation [32,52]. In addition, this error also occurs because the FEM results that are used as reference values will be affected by the quality and size of the mesh.

The deformation area (see Figure 9b) is roughly triangular in shape, which is complementary to the beam shape. To describe the resonance frequency characteristics of the amplifying mechanism accurately by weakening the stress concentration effects, we constructed a hybrid flexibility-modified model of the amplifying mechanism. Specific measures were applied to establish a stress concentration transfer matrix in this area and then add it to the total transfer equation to correct the equation results. A triangular deformation area was added into the beam deformation area. Therefore, the stress concentration transfer matrix and the beam transfer matrix can be combined to give a new modified beam transfer matrix. The length of the modified beam is:(5)$$L_{new}=L_{beam}+\Delta L$$
where ΔL=f(θ,b,...). The functional relationship can be obtained through numerical fitting, and its independent variables should include the angle, thickness, and connection width of the pseudo-rigid body element of the bridge arm, etc. [53,54,55]. Owing to the main parameters affecting the accuracy of the calculation results being angle and thickness, this article only conducted numerical fitting for these two cases.

Comparison results for the natural frequencies of the improved model and those of the model before improvement are shown in Figure 10. Figure 10a shows the corrected curves of amplifying mechanisms with different bridge arm thicknesses under linear fitting correction (ΔL=K×b). Figure 10b illustrates the calibration curves of amplification mechanisms with different bridge arm angles under the correction of polynomial fitting functions ($\Delta L=K_{1}\times \theta^{3}+K_{2}\times \theta^{2}+K_{3}\times \theta +K_{4}$). The results show that the modified model can describe the dynamic and static performance of the mechanism more accurately than the mathematical models before the modification.

## 6. Iterative Optimization

From the results of the characteristic analyses described in the previous sections, we have gained a clear understanding of the main parameters that affect the amplification ratio and the resonance frequency of the amplifying mechanism. However, these two characteristics are contradictory when used in selection of the bridge arm thickness, as can be seen from Figure 4b and Figure 6a. Therefore, it is impossible to design a mechanism that has the two best values for these characteristics simultaneously, and the thickness and angle of the bridge arm can only be selected according to specific design requirements [56].

Using the MSTMM, we can establish a functional relationship between the dynamic and static characteristics and the main influencing parameters as follows:(6)$$A_{amp}=f\left(\theta ,b,b_{i},b_{o},...\right)$$
(7)$$F_{res}=f\left(\theta ,b,b_{i},b_{o},...\right)$$

On this basis, by performing iterative parameter optimization, the parameter values that meet the design requirements can be found and the optimization model is then designed [51]. The optimization flowchart is shown in Figure 11 (note that only selected angle and thickness parameters are given here).

An output displacement of more than 300 µm and a main resonance frequency of more than 300 Hz were selected as the design goals (the piezoelectric actuator used is the Physik Instrumente’s (Karlsruhe, Germany) product P-888.91 [57]). While considering of the effects of the calculation errors and the stress concentration, the structural parameter values with the best characteristic result values should be selected as far as possible during this process. Here, 80% of the calculated result of the output displacement at the end of the amplifying mechanism is taken as the result value. Figure 12 shows the values of all main parameters that meet the design requirements.

One set of results was processed up to the design stage. The specific parameters are listed in Table 1.

According to the design parameters in Table 1, the MSTMM correction mathematical model was established. The calculated output displacement of the mechanism is 330.77 µm, and the resonance frequency in the main direction is 340 Hz.

## 7. Experiment

To verify the feasibility of the proposed model, the rhombus-type compliant mechanisms model were processed by wire cutting. The mechanisms were driven by piezoelectric ceramic actuators produced by Physik Instrumente, and the specific actuator parameters are given in Table 2. The output displacement of the mechanism was measured using a laser displacement meter (CL-P070, Keyence, Osaka, Japan) with a resolution of 0.25 µm.

The experimental setup is pictured in Figure 13. By inputting a voltage to the piezoelectric stack, the mechanism produces an output displacement and thus allows the static characteristics and the dynamic characteristics of the rhombus-type amplifying mechanism to be obtained [58].

First, the hysteresis curve of the amplification mechanism is measured, as shown in Figure 14a. Second, the maximum output displacement of the mechanism is tested (see the results in Table 3). It can be seen that the maximum output displacement of the amplifying mechanism is 336.5 µm. The error of see Figure 14b (FEM) and MSTMM results is within 2%. Third, Figure 14c shows the frequency sweeping results obtained for the mechanism. The result shows that the resonance frequency of the mechanism in the main direction is 334.8 Hz, with errors of approximately 3% and 1% compared to the FEM and MSTMM results, respectively. Compared with the literature [23,31,32,37], the results are closer to reality and more accurate than traditional mathematics models.

Strong similarity was observed between the results of the finite element analysis and those of the proposed model, and the deviations appeared to be caused by manufacturing and assembly errors of the amplifying mechanism, the installation of the piezo-stacks, and sensor errors when affected by measurement noise, light interference, vibration, and other factors [23,33].

## 8. Application

The rhombus-type mechanism analyzed in this article can be utilized for applications where large strokes and high natural frequencies are required. An example of a possible application can be found in focusing mechanisms(see Figure 15a). The rhombus-type amplifying mechanism mainly amplifies the displacement of the actuators, and a thin and long flexible beam is attached to it to limit the offset of the amplifying mechanism in the non-working direction(see Figure 15b), so as to realize the translation of the focusing mechanism in the one-dimensional direction [59,60,61,62].

Based on the analysis method proposed above, the mathematical model of the one-dimensional focusing mechanism is established. Next, with the goal of large stroke and high natural frequency, the main design dimensions of the focusing mechanism are optimized using the iterative optimization method proposed in Section 6. The specific parameter indicators of the focusing mechanism are listed in Table 4.

The model is processed by wire-cutting technology. Furthermore, the material selection is Spring steel 65 Mn. A simple test was carried out on the model to verify the feasibility of using the rhombus-type mechanism in the focusing mechanism. The output displacement is measured with a laser displacement meter, and when the PZA is at 120 V, a maximum output displacement of about 450 µm is produced (see Figure 16b). Furthermore, the frequency response [63] is measured with the frequency response meter, and the resonance peak is near 302 Hz (see Figure 16c). Other parameters are shown in Table 5.

In this case, the feasibility of determining the optimal values for the mechanism’s parameters based on the MSTMM approach is verified, and certain reference values are provided for design of the precision focusing mechanism.

## 9. Discussion and Conclusions

In this paper, a parameterized dynamic and static mathematical model of a compliant-amplifying mechanism has been established based on the multi-body system transfer matrix method. On this basis, the effects of the main parameters of the mechanism on the static characteristics of the displacement amplification ratio and the dynamic characteristics of the resonance frequency in the model were analyzed using the control variable method. The transfer matrix was corrected by adding correction coefficients that reduced the influence of stress concentration. Finally, the parameter values were largely optimized to meet the mechanism’s design requirements. Based on the discussion above, the main contributions of this paper can be summarized as follows:(1)A unified dynamic and static mathematical model of the compliant-amplifying mechanism is established;(2)A comprehensive analysis of the effects of all geometric parameters and material properties on the dynamic and static characteristics of the mechanism has been performed;(3)The amplification ratio decreases with increasing bridge arm angle following a peak value, and decreases with decreasing arm thickness and length. Greater bridge arm thickness leads to a smaller arm angle, a longer output rigid body, and a higher first-order resonance frequency. Larger values for the length and height of the mechanism’s input rigid body and a larger value for the height of the output rigid body cause the first-order resonance frequency to decrease;(4)The revised mathematical model has been established to reduce errors caused by stress concentration. Through experimental verification, the error between the two is maintained within 2%. The mathematical model is closer to the actual situation, and the calculation accuracy is better than the traditional mathematical model;(5)An optimal rhombus-type amplifying mechanism model and a spatial focusing mechanism were designed using the iterative optimization method, which has a certain reference value for use in engineering applications.

It was thus proven that it is feasible to use the mathematical model of the compliant amplification mechanism based on the MSTMM to describe the mechanism’s dynamic and static characteristics using the analysis above.

## Figures and Tables

**Figure 1 micromachines-14-01502-f001:**
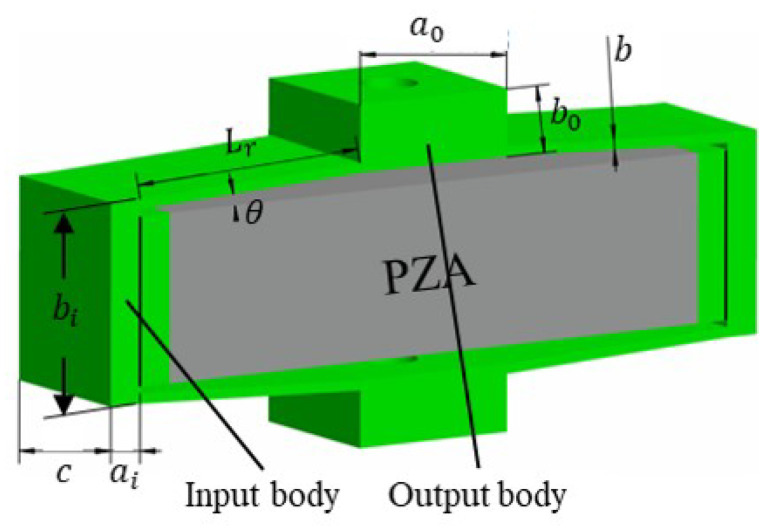
Parametric models of compliant mechanisms.

**Figure 2 micromachines-14-01502-f002:**
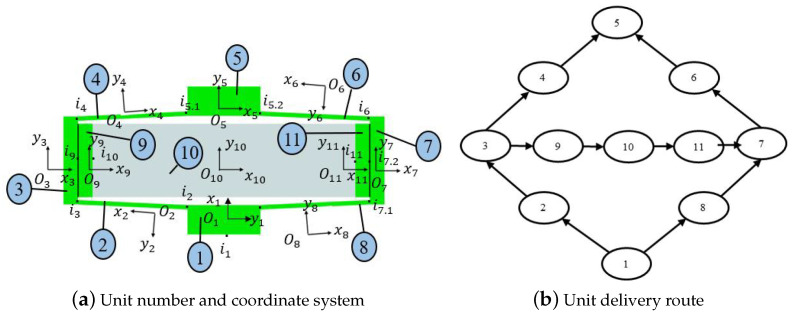
Topology diagram of compliant-amplifying mechanism.

**Figure 3 micromachines-14-01502-f003:**
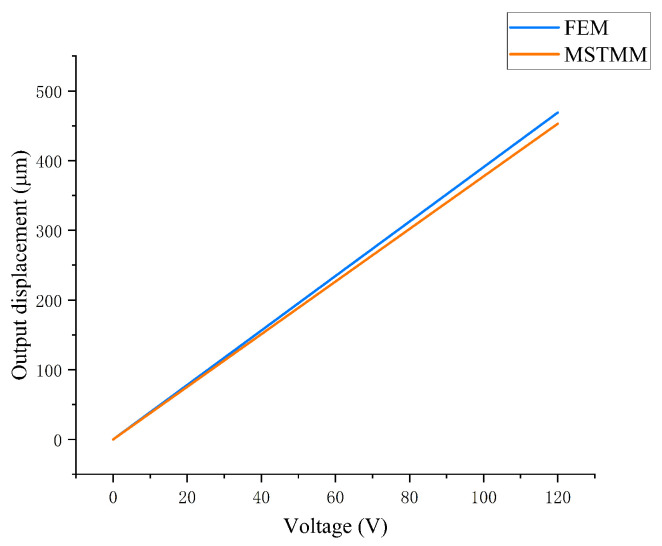
Relationships between the output displacement of the mechanism and the piezoelectric input voltage for the FEM and MSTMM models.

**Figure 4 micromachines-14-01502-f004:**
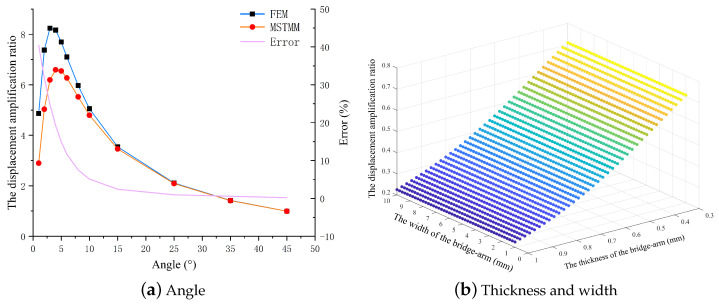
Effects of bridge arm parameters on displacement amplification ratio.

**Figure 5 micromachines-14-01502-f005:**
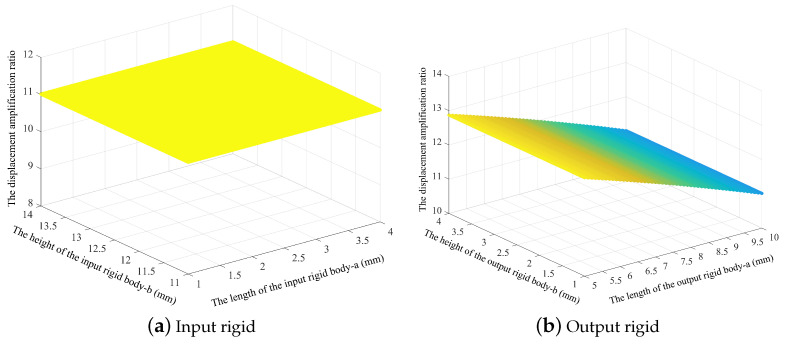
Effects of size of the input and output pseudo-rigid body units of the amplifying mechanisms on the displacement amplification ratio.

**Figure 6 micromachines-14-01502-f006:**
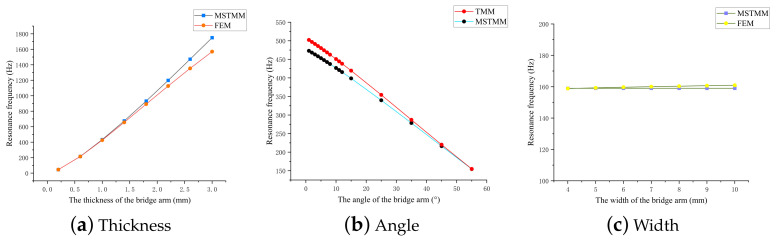
Effects of the main parameters of the amplifying mechanism on the first-order resonance frequency.

**Figure 7 micromachines-14-01502-f007:**
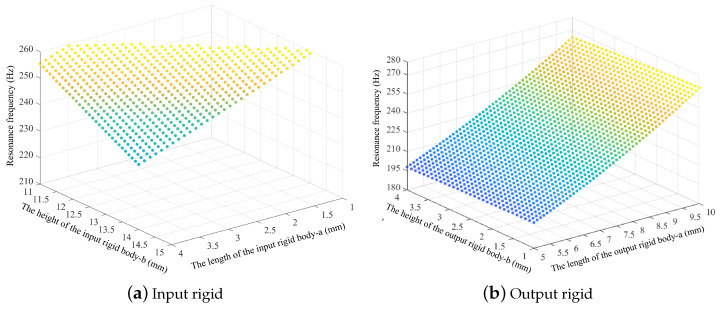
Effects of the main geometric parameters of the pseudo-rigid units on the resonance frequency.

**Figure 8 micromachines-14-01502-f008:**
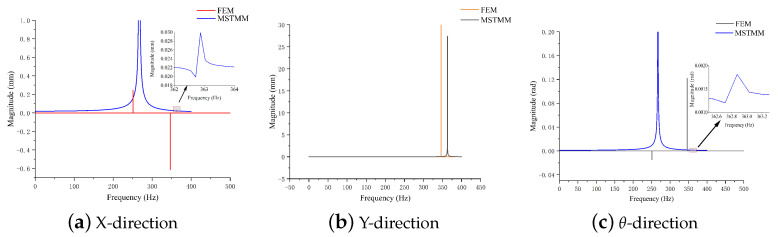
Mechanism frequency response curves for FEM and MSTMM models.

**Figure 9 micromachines-14-01502-f009:**
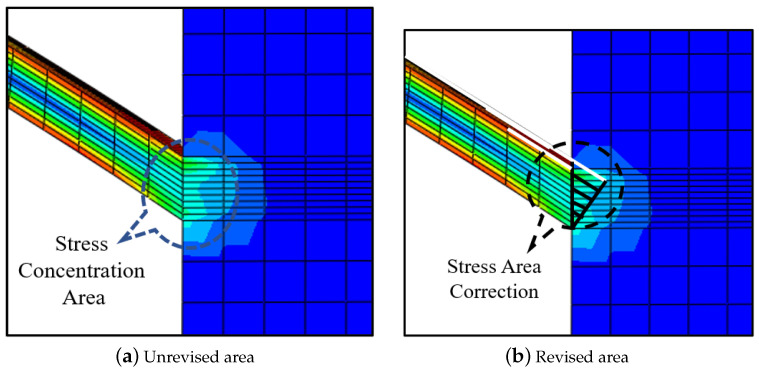
Stress concentration area and stress correction area.

**Figure 10 micromachines-14-01502-f010:**
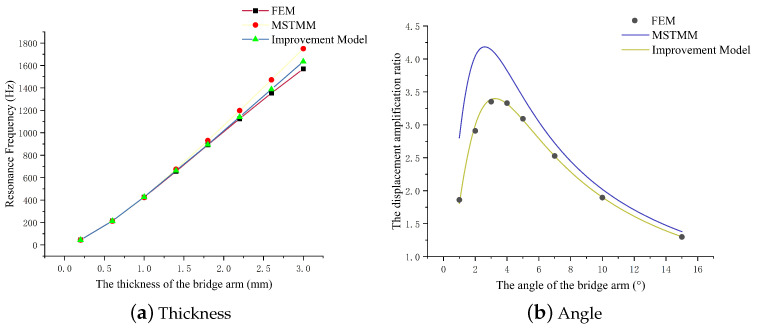
Corrected model curves.

**Figure 11 micromachines-14-01502-f011:**
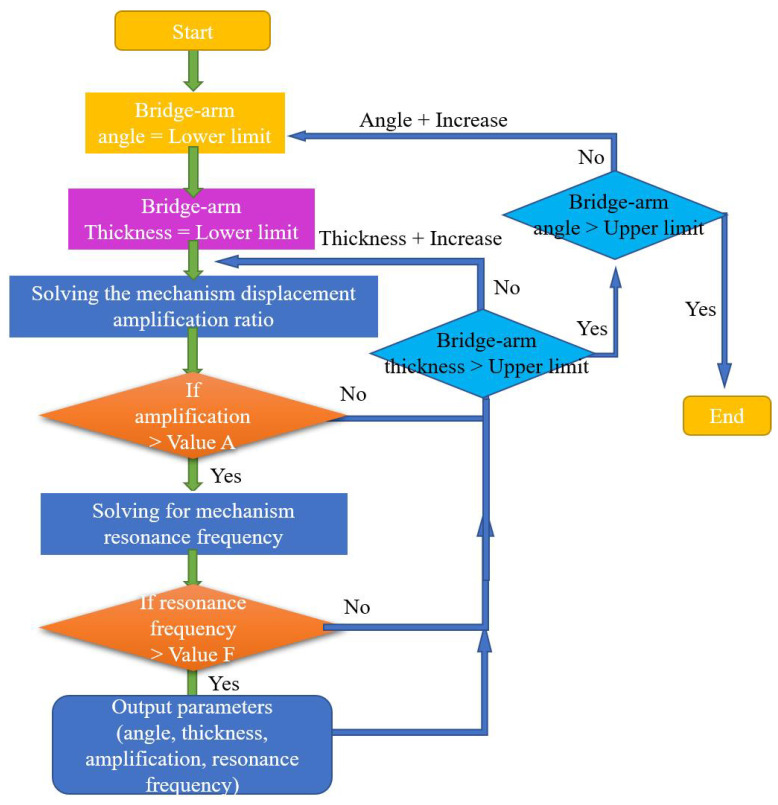
Iterative optimization process flowchart.

**Figure 12 micromachines-14-01502-f012:**
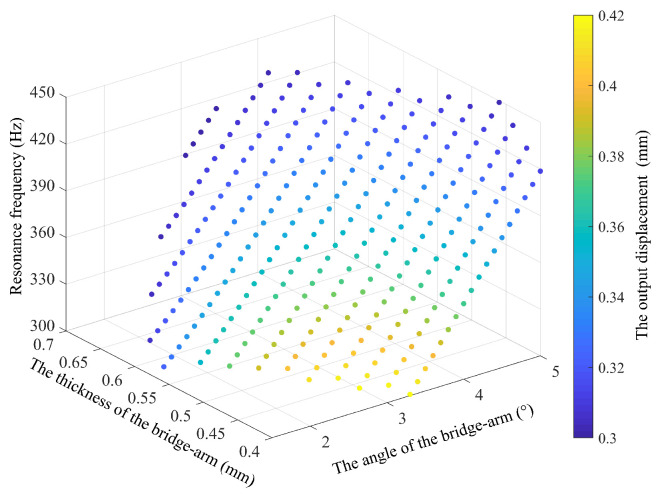
Iterative optimization results.

**Figure 13 micromachines-14-01502-f013:**
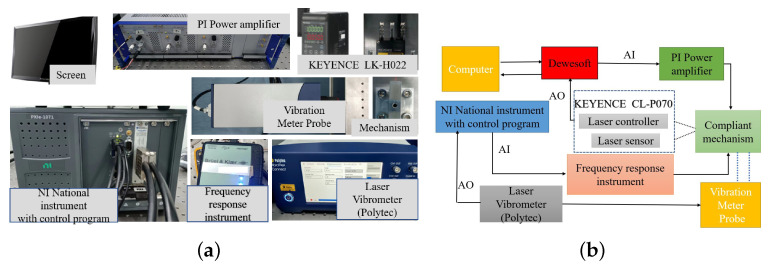
(**a**) Experimental setup and (**b**) schematic diagram of the overall operation experimental system.

**Figure 14 micromachines-14-01502-f014:**
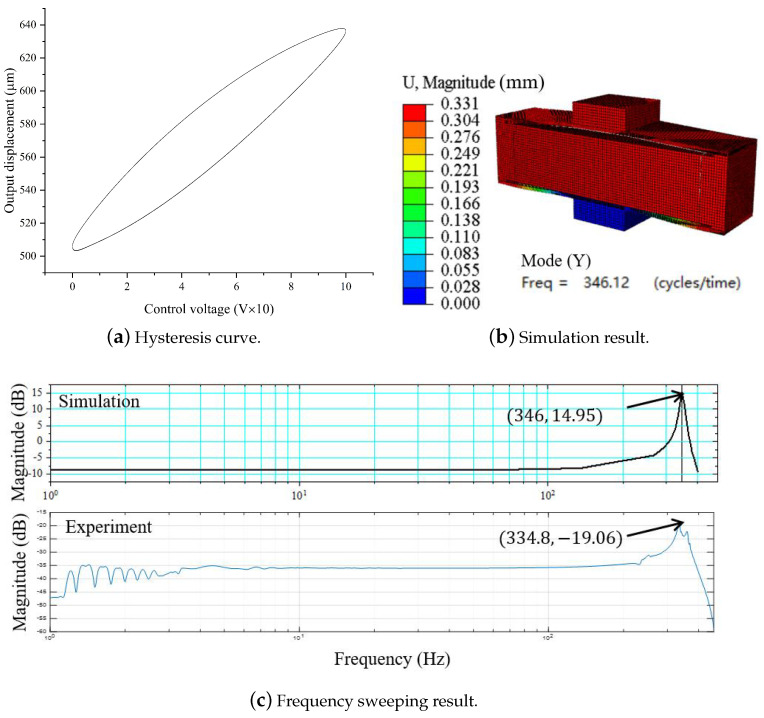
Experimental and simulation results of rhombus-type amplifying mechanism.

**Figure 15 micromachines-14-01502-f015:**
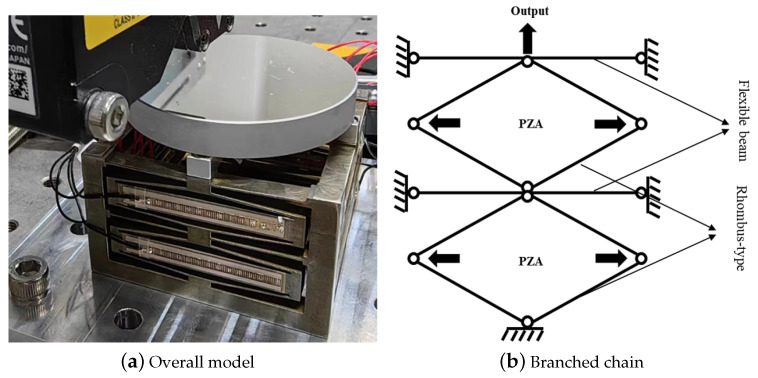
Precision focusing mechanism.

**Figure 16 micromachines-14-01502-f016:**
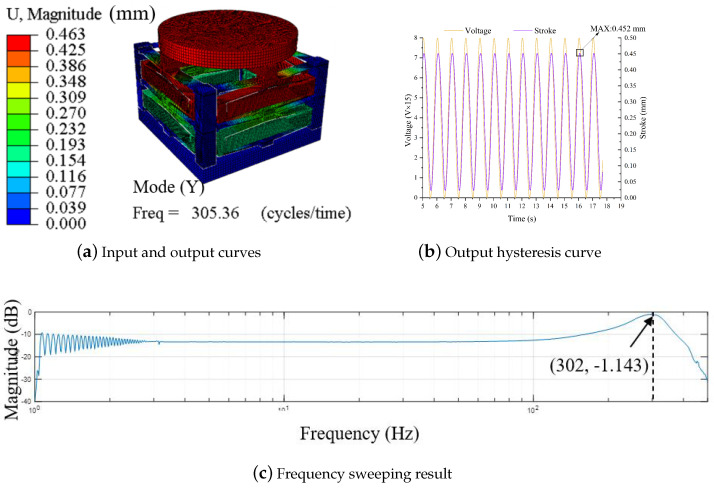
Simulation and experimental results for the focusing mechanism.

**Table 1 micromachines-14-01502-t001:** Specific parameters of the amplifying mechanism.

$L_{r}$ (mm)	*c* (mm)	$\theta \;{(}^{\circ })$	*b* (mm)	$a_{o}$ (mm)	$b_{o}$ (mm)
-	10	2.6	0.6	10	4
$a_{i}$ (mm)	$b_{i}$ (mm)	Material	*E* (MPa)	μ	ρ (g/cm${}^{3}$)
2	12	Al7075	71,000	0.33	2.81

*E* is the elastic modulus. μ is the Poisson’s ratio. ρ is the density of the material. The meanings of the parameters in the following table are the same as this.

**Table 2 micromachines-14-01502-t002:** Main parameters of the piezoelectric actuator.

Dimensions (mm × mm × mm)	Maximum Travel Range (µm)	ρ (g/cm${}^{3}$)	μ	*E* (MPa)
10×10×36	38±15%	7.5	0.3	36,000

**Table 3 micromachines-14-01502-t003:** Maximum output displacement test results for the mechanism.

Times	1	2	3	Avg.
Stroke (µm)	336.75	336	337.75	336.5

**Table 4 micromachines-14-01502-t004:** Specific parameter indicators of the focusing mechanism.

Performance	Value	Unit
Stroke	>420	µm
Resonance frequency	>300	Hz
Height	<50	mm
Length	<60	mm
Width	<60	mm
Mass	<200	g

**Table 5 micromachines-14-01502-t005:** Other parameters of the focusing mechanism.

Height (mm)	Length (mm)	Width (mm)	Mass (g)
44	54	54	195

## Data Availability

Not applicable.

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
