# Peer review of "Analysis and Optimization of Dynamic and Static Characteristics of the Compliant-Amplifying Mechanisms"

_micromachines, 2023, doi:10.3390/mi14081502_

Round 1

Reviewer 1 Report

The authors propose a comprehensive theoretical model of compliant amplifying mechanisms based on the multi-body system transfer matrix method to analyze the dynamic and static characteristics of compliant amplifying mechanisms. There are a few comments needed to be addressed.

1. FEM should be a much clearer method and easily to study the effects of geometric parameters. It is easy to change the geometric parameters in the FEM simulation model. Also, the results are 3D and much clearer. What are the advantages of mathematical method?

2. Less introduction and details of MSTMM are included. I suggest to include basic introduction of MSTMM for a better readership.

3. Figure 14 is not clear. Please revise.

4. There are 25 figures in total. Could the authors integrate a few figures together to reduce the total number of figures?

5. Also, from Figure 14, the FEM simulation results have big differences compared to the MSTMM results. Why?

Minor editing of English language required.

Reviewer 2 Report

Dear Authors,

My comments about your manuscript are listed in the attachment. 

Kind Regards

English should be polished

Round 2

Reviewer 2 Report

Dear Authors,

The corrections were performed. I have no futher questions or remarks.

The manuscript can be published. 

Kind Regards 

Author Response

Thanks for your suggestion.We tried our best to improve the manuscript , and hope that the correction will meet with approval.